



# Vertical cloud radiative heating in the tropics: Confronting the EC-Earth model with satellite observations

Erik Johansson[1,2,3], Abhay Devasthale[1], Michael Tjernström[2,3], Annica M. L. Ekman[2,3], Klaus Wyser[4], and Tristan L'Ecuyer[5]

[1]Atmospheric Remote Sensing, Research and development department, Swedish Meteorological and Hydrological Institute (SMHI), Norrköping, Sweden.
[2]Department of Meteorology, Stockholm University (MISU), Stockholm, Sweden.
[3]Bolin Center for Climate Research, Stockholm University, Stockholm, Sweden.
[4]Rossby Centre, Research and development department, Swedish Meteorological and Hydrological Institute (SMHI), Norrköping, Sweden.
[5]Department of Atmospheric and Oceanic Sciences, University of Wisconsin-Madison, Madison, USA.

**Correspondence:** Erik Johansson (erik.johansson@smhi.se)

**Abstract.** Understanding the coupling of clouds to large-scale circulation is one of the grand challenges for the global climate research community. In this context, realistically modelling the vertical structure of cloud radiative heating/cooling (CRH) in Earth system models is a key premise to understand these couplings. Here, we evaluate CRH in two versions of the European Community Earth System Model (EC-Earth) using retrievals derived from the combined radar and lidar data from the CloudSat and CALIPSO satellites. One model version is also used with two different horizontal resolutions. Our study evaluates large-scale intraseasonal variability in the vertical structure of CRH and cloud properties and investigates the changes in CRH during different phases of the El Niño Southern Oscillation (ENSO), a process that dominates the interannual climate variability in the tropics.

EC-Earth generally captures both the intraseasonal and meridional pattern of variability in CRH over the convectively active and stratocumulus regions and the CRH during the positive and negative phases of ENSO. However, two key differences between model simulations and satellite retrievals emerge. First, the magnitude of CRH over the convectively active zones is up to twice as large in the models compared to the satellite data. Further dissection of net CRH into its shortwave and longwave components reveals noticeable differences in their vertical structure. The shortwave component of the radiative heating is overestimated by all model versions in the lowermost troposphere and underestimated in the middle troposphere. These over- and underestimates of shortwave heating are partly compensated by an overestimate of longwave cooling in the lowermost troposphere and heating in the middle troposphere. The biases in CRH can be traced back to disagreements in cloud amount and cloud water content. There is no noticeable improvement of CRH by increasing the horizontal resolution in the model alone. Our findings highlight the importance of evaluating models with satellite observations that resolve the vertical structure of clouds and cloud properties.



## 1 Introduction

About 70% of Earth is on average covered by clouds (King et al., 2013; Karlsson and Devasthale, 2018; Stubenrauch et al., 2013). Clouds are, however, unevenly distributed in time and space, particularly in the tropics. Regions with marine stratocumulus clouds or with persistent convection, such as the intertropical convergence zone, can reach up to 90% cloud cover, while over deserts and parts of the open ocean, cloud cover can be as low as 20% when averaged over time.

In addition to playing an essential role in the hydrological cycle, clouds are crucial in regulating Earth's radiative balance (Stephens, 2005). By reflecting incoming solar (shortwave) radiation, they increase the planetary albedo, contributing to a cooling of Earth. At the same time, clouds absorb thermal (longwave) radiation and re-emit energy to space from cloud-tops at a lower temperature than the surface, contributing to a warming of Earth. The absorption and emission of longwave radiation of clouds, and also of different atmospheric gases, are usually referred to as the greenhouse effect. The net result of radiation interaction with clouds varies strongly with time, location, and cloud type.

The distribution of heating from clouds in space and time is an essential driver of circulation in the ocean and the atmosphere (e.g. Slingo and Slingo, 1988; Randall et al., 1989; Sherwood et al., 1994). Meridional differences in cloud radiative forcing, where tropical clouds have a stronger positive forcing compared to mid-latitude clouds, enhance meridional temperature gradient and therefore impact the Hadley circulation (Sherwood et al., 1994; Raymond, 2000). Similarly, zonal differences in the tropics between convective and non-convective zones impact the zonal overturning eddies, such as the Walker circulation (Sherwood et al., 1994). In the tropics, the CRH can influence troposphere-to-stratosphere transport (Corti et al., 2006; Johansson et al., 2019). On a smaller scale, radiative heating can alter temperature profiles inside clouds, hence, creating buoyancy-driven turbulence within the cloud (Ackerman et al., 1988; Wood, 2012).

El Niño – Southern Oscillation, or ENSO, is the major source of interannual climate variability in the tropics. This variability is due to variations in the strength of the easterly trade winds, known as the Walker circulation, that modify the sea surface temperature (SST) pattern across the Pacific, leading to a zone of strong convection over the West Pacific. At the same time, marine stratocumulus clouds will prevail over the East Pacific. During an El Niño, trade winds weaken, or even reverse, with a significant effect on the cloud distribution over the Pacific. Furthermore, the effects of ENSO have been shown to spread far beyond the Pacific Ocean (Madenach et al., 2019, and references within). The opposite, when the trade winds are enhanced, is usually referred to as La Niña.

Clouds and radiation are described differently in different climate models. However, they all have in common that they need some sort of parameterization to resolve the sub-grid structure of clouds. Inadequacies in how this is done increase uncertainty of the modelled climate. Exarchou et al. (2017) found that unrealistically low cloud cover in EC-Earth creates a warm SST bias over the Tropical Atlantic. The same study also found that an increase in horizontal resolution did not reduce this SST bias. On the other hand, Hourdin et al. (2013) found that an increase in horizontal resolution in the IPSL-CM5A coupled model led to an improved cloud cover. This ambiguousness can also be found in Prodhomme et al. (2016), who found that an increased horizontal resolution in an earlier version of EC-Earth improved the representation of the Indian monsoon while there was no improvement for the African monsoon.





The importance of realistic observation data for model development, evaluation and tuning have been pointed out in several
studies (McFarlane et al., 2007; Mauritsen et al., 2012; Bojinski et al., 2014). Traditionally, observation datasets have either
had a poor spatial resolution, for example, data from field campaigns, or no or insufficient vertical information, such as data
from many passive satellite sensors. Therefore, the evaluation of radiation in models usually focuses on the top of the atmo-
sphere (TOA) (Thomas et al., 2019). This lack of observations was dramatically altered with the deployment of CloudSat and
CALIPSO (Cloud-Aerosol Lidar and Infrared Pathfinder Satellite Observations) satellites. The cloud radar on-board CloudSat
can provide detailed information on the vertical structure of clouds while CALIPSO detects even sub-visual cirrus clouds.

This study aims to compare the vertical structure of clouds and cloud radiative heating (CRH) in the tropics from the global
climate model EC-Earth with a combined dataset from the CloudSat and CALIPSO satellites sensors. The study will focus on
variability across both seasons and ENSO events. Two different versions of EC-Earth are used and for one of these, we also
employ two different horizontal resolutions; the default resolution and one higher to explore how resolution affects the CRH.

## 2    Data and Method

### 2.1    Model

The atmospheric model of EC-Earth v3 is based on the Integrated Forecasting System cycle 36r4 (IFS CY36r4) from the Euro-
pean Centre for Medium-Range Weather Forecast (ECMWF). We use two different versions of EC-Earth v3; the first was used
in the PRocess-based climate sIMulation: AdVances in high-resolution modeling and European climate Risk Assessment (PRI-
MAVERA) project, while the second, v3.3.1, was used for phase 6 of the Coupled Model Intercomparison Project (CMIP6).
EC-Earth v3 is a continuation of EC-Earth v2.3 used in CMIP5 (Hazeleger et al., 2012, 2013). Compared to v2.3, EC-Earth v3
has an updated microphysics scheme to, among other things, better represent mixed-phase clouds (Forbes et al., 2011). There
is also a new convection scheme which increases the intraseasonal variability, leading to more intense convection over the con-
tinents (Bechtold et al., 2008) and a better representation of the daily convective cycle (Bechtold et al., 2013). Together with a
new McRad radiation scheme the convection scheme also allows for a more detailed representation of the interaction between
radiation and sub-grid clouds (Morcrette et al., 2008). Furthermore, v3.3.1 uses a new surface/vegetation albedo scheme and
an improved description of the indirect aerosol effect compared to the PRIMAVERA version.

For the PRIMAVERA version of EC-Earth, we use two different horizontal resolutions, T511L91 and T255L91. T511L91
is a high-resolution version, with a horizontal resolution of $0.35°$, i.e., $\sim 40km$ at the equator, while the T255L91 is the
standard-resolution with a horizontal resolution of $0.70°$, i.e., $\sim 80km$ at the equator. EC-Earth v3.3.1 is only used with the
standard-resolution T255L91. All versions use 91 vertical levels. The vertical levels are not equally distributed throughout the
atmosphere. Instead, resolution is highest in the lower part of the atmosphere and decreases as the altitude increases. However,
to fit with the vertical resolution of the satellite observations, the output is interpolated by the model to a constant $480m$
vertical resolution. Since one focus of this study is on the horizontal resolution, the time step for all model versions is set to
$900s$, i.e., the time step usually used for T511L91. The model output is every third hour. The model simulations are performed
as atmosphere only, i.e. with prescribed SSTs. The simulations with the PRIMAVERA versions use daily SSTs interpolated to





the different resolutions, while v3.3.1 uses monthly SSTs. For brevity the high-resolution PRIMAVERA version is henceforth called EC-Earth3P-HR, the standard-resolution PRIMAVERA is called EC-Earth3P, and v3.3.1 used for CMIP6 is referred to as EC-Earth3.

## 2.2 Satellite Observations

The satellite observations used in the study combine observations mainly from CloudSat and CALIPSO. CloudSat carries a $94GHz$ cloud profiling radar (CPR) (Stephens et al., 2008) and CALIPSO carries the Cloud-Aerosol Lidar with Orthogonal Polarization (CALIOP) measuring at $532$ and $1064nm$ (Winker et al., 2009). They were both deployed in 2006 as part of the A-Train constellation (Stephens et al., 2002; L'Ecuyer and Jiang, 2010), where all satellites travel along the same path within just seconds to minutes. This setup allows for a dataset where measurements from instruments onboard different satellites easily can be combined. The 2B-FLXHR-LIDAR product (L'Ecuyer et al., 2008; Henderson et al., 2013) provides calculated radiative properties, the 2B-CLDCLASS-LIDAR product (Sassen et al., 2008) provides the vertical cloud mask, and 2B-CWC-RVOD (Austin et al., 2009) provides the cloud ice and liquid water contents (IWC and LWC). These datasets have been used extensively for cloud studies in the tropics (e.g. Johansson et al., 2015; Matus and L'Ecuyer, 2017; Hartmann and Berry, 2017; L'Ecuyer et al., 2019; Hang et al., 2019).

The 2B-FLXHR-LIDAR product uses a two-stream radiative transfer model with twelve longwave bands and six shortwave bands. The state variables needed, e.g., temperature and specific humidity profiles, are obtained for each CloudSat profile from ECMWF reanalysis data, available from the ECMWF-AUX dataset. Various CloudSat and CALIPSO products provide cloud and aerosol properties. CloudSat cannot provide direct information on the phase of the cloud hydrometeors. Instead, this retrieval relies on temperature profiles from the ECMWF-AUX dataset together with data from the MODIS instrument onboard Aqua, which introduces a large uncertainty in the IWC/IWP data (Devasthale and Thomas, 2012).

## 2.3 Method

To evaluate the vertical structure of clouds in the EC-Earth model, and the corresponding CRH, we use data from four years of CloudSat and CALIPSO data, January 2007 to December 2010. The model simulations start in 2005, and all model versions were run for six years until 2010; 2005 and 2006 are considered as spin-up time and are excluded from the analysis. Furthermore, December 2009 is also excluded due to a lack of satellite data for this month. Four years is not enough to investigate interannual variability, but should be sufficient for carrying out a statistical evaluation considering the AMIP-type simulations. We focus on the tropical region ($30°S$ and $30°N$) and evaluate the four seasons, Dec–Feb, Mar–May, Jun–Aug and Sep–Nov, separately.

Note that the satellite simulator for evaluating heating rates in EC-Earth is currently not available. CloudSat and CALIPSO pass the equator at roughly 13:30 local time during daytime, so the model results are linearly interpolated from the two nearest output times to fit the satellite overpass time. This time interpolation together with the ability of the active satellite instruments to detect thin clouds reduces the need for satellite simulator otherwise commonly used for passive instruments (see, e.g. Pincus et al., 2012).





120    The satellite data is projected on a $1°$ by $1°$ longitude/latitude grid while retaining the native $240m$ vertical resolution. Due to ground clutter contamination in the CloudSat data, the lowest $750m$ are excluded. When the differences between the different data and model outputs are calculated, the dataset with higher resolution is averaged to the lower resolution. For both the satellite retrievals and the model output, the CRH is calculated by subtracting the shortwave and longwave clear sky values from the cloudy sky values (in models typically referred to as "all-sky") (Johansson et al., 2015).

$$CRH = ((SWHR_{cloudy} + LWHR_{cloudy}) - (SWHR_{clear} + LWHR_{clear})) \qquad (1)$$

where $SWHR$ is the shortwave heating rate and $LWHR$ is the longwave heating rate. A two-sided student t-test is used on the monthly mean values and the $95\%$ confidence interval is highlighted with black dots in the figures. We also did a Mann-Whitney rank test with similar results as the student t-test and these results are therefore not shown. Even so, the small sample size, especially for the ENSO study, will limit the interpretation of the statistical test.

130    The cloud water content in EC-Earth is given in $kgkg^{-1}$, while the satellite dataset provide the water content in $gm^{-3}$. We therefore need to recalculate the water content from the model by multiplying with density

$$WC[gm^{-3}] = WC[kgkg^{-1}] * \rho \qquad (2)$$

where $WC$ is the cloud water content (either ice or liquid) and $\rho$ is the density. $\rho$ is calculated from the ideal gas law (Holton and Hakim, 2012, chapter 2.9)

$$\rho = p/(R_d * T_v) \qquad (3)$$

and

$$T_v = T(1 + (R_v/R_d - 1) * q_v) \qquad (4)$$

where $p$ is the pressure, $T_v$ is the virtual temperature, $T$ is the temperature, $q_v$ is the specific humidity, and $R_d$ and $R_v$ is the gas constant for dry air and water vapour. We use the constants $287.04 Jkg^{-1}K^{-1}$ and $461 Jkg^{-1}K^{-1}$ for $R_d$ and $R_v$ respectively while the model is providing the other variables for each vertical level.

    The phase of the ENSO is obtained from the National Oceanic and Atmospheric Administration (NOAA; see (https://origin.cpc.ncep.noaa.gov/products/analysis_monitoring/ensostuff/ONI_v5.php). The ENSO index is based on SST anomalies over the Niño 3.4 region ($5°N$–$5°S$, $120°W$–$170°W$). The phases are based on a threshold of $+0.5°C$ ($-0.5°C$) for positive (negative) ENSO relative to a 30-year period. The two phases are henceforth referred to as ENSOP (positive) and ENSON (negative). The months with positive ENSO (10 months) are averaged for the ENSOP analysis, and similarly for the month with ENSON (22 months). The mean value for all months during the four years is then subtracted from the ENSOP and ENSON to create anomalies. Since the impact of ENSO is largest close to the equator, we only use model and satellite data between $15°S$ and $15°N$ for this analyse.

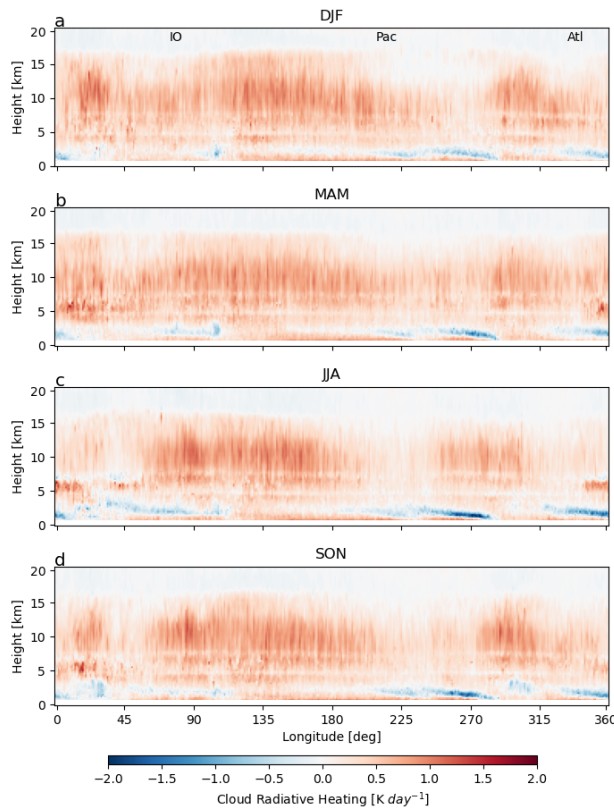

**Figure 1.** The vertical structure of cloud radiative heating based on the 2B-FLXHR-LIDAR dataset averaged between latitude $\pm 30°$. IO = Indian Ocean, Pac = Pacific Ocean and Atl = Atlantic Ocean. The longitude bins are spaced at $1°$. a) December–February (DJF), b) Mars–May (MAM), c) June–August (JJA) and d) September–November (SON).

## 3 Results

### 3.1 Seasonal vertical cloud radiative heating

#### 3.1.1 Meridional average

In this section we investigate the seasonal vertical CRH in the tropics. Figures 1–4 all show average values, or average differences, over $\pm 30°$ latitude divided into the four seasons. Figure 1 shows the vertical structure of CRH derived from the satellite observations. During all seasons, clouds have a net heating effect in the upper troposphere, from 5 to $10–15 km$. Close to the equator, the intertropical convergence zone constitutes the rising branch of the Hadley Cell. This zone has a prevalent cloud cover with small seasonal variations within the $\pm 30°$ latitude band and will, therefore, contribute to CRH throughout the year. Over the warm tropical waters, large convective systems exist and these clouds will generate strong CRH. This CRH is especially pronounced at higher altitudes during seasons when the convection is most active. The highest ocean surface temper-



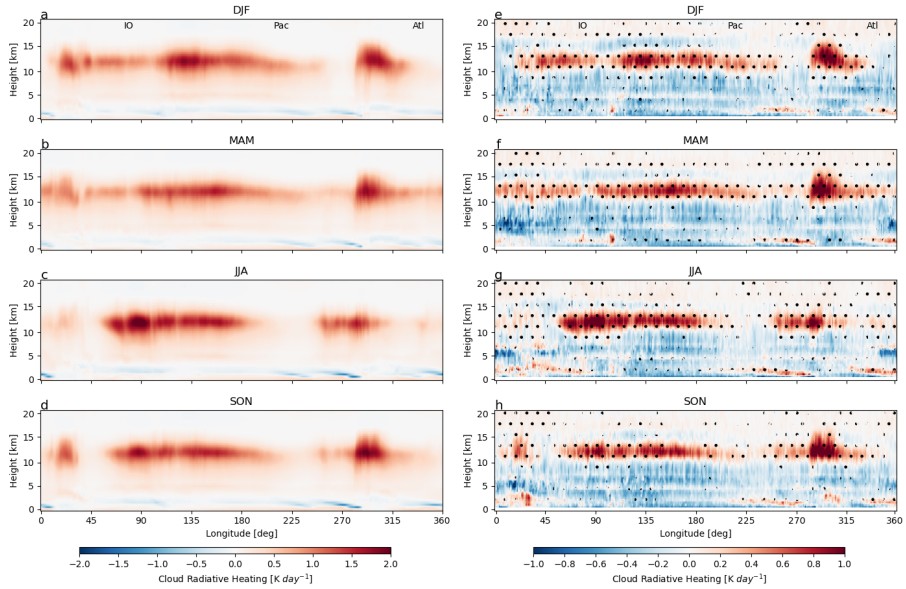

**Figure 2.** As in Figure 1, but showing (a–d) the vertical structure of cloud radiative heating from the high-resolution EC-Earth (EC-Earth3P-HR) and (e–h) the difference in cloud radiative heating, between satellite observations (Figure 1) and EC-Earth3P-HR (EC-Earth3P-HR minus satellite). Statistically significant differences, at the 95% confidence using a student t-test based on monthly mean values, are marked with black dots (e–h).

atures are found in the western Pacific and eastern Indian Ocean (sometimes referred to as the tropical warm pool; 70°–160°)
and the convective cloud systems here contribute to the radiative heating ($1.5 K day^{-1}$) with a small seasonal cycle. In DJF and MAM, during the rainy period over south-central Africa (340°–45°), there is strong CRH ($1.5 K day^{-1}$ and $1 K day^{-1}$, respectively) compared to $0.5 K day^{-1}$ during the dry season (JJA). The signature of the South Asian monsoon ($\sim 90°$) is also visible in the CRH ($1.5 K day^{-1}$) during JJA and SON (also see Johansson et al., 2015). During the SON months, the intense but local CRH ($1.5 K day^{-1}$) over south-central America (280°–315°) also stands out.

West of the continents, cold upwelling water creates vast areas with marine stratocumulus clouds that are present on average 60% of the time. These clouds generate the sloping wedges of low-level radiative cooling seen in the lower troposphere, mostly below $2 km$, with maxima sloping down towards the west coast of the continents at low altitude. Marine stratocumuli have a seasonal cycle with peaks in cloud fraction in August – October (Hahn and Warren, 2007, updated 2009), leading to intense cooling ($-1.5 - -2 K day^{-1}$) in JJA and SON close to South America and Africa (see Figures 1c–d).

We now explore if the high-resolution version of EC-Earth can simulate the large-scale CRH features retrieved from satellite data. Similar to Figure 1, Figures 2(a–d) shows the CRH as it is simulated in EC-Earth3P-HR, while Figures 2(e–h) shows the differences between the simulation and the satellite observations.

EC-Earth3P-HR captures the pattern of both meridional and seasonal variability in CRH, visible in the satellite-retrieved data. For example, the pronounced CRH rates over the warm pool region, over south-central America during the SON months,



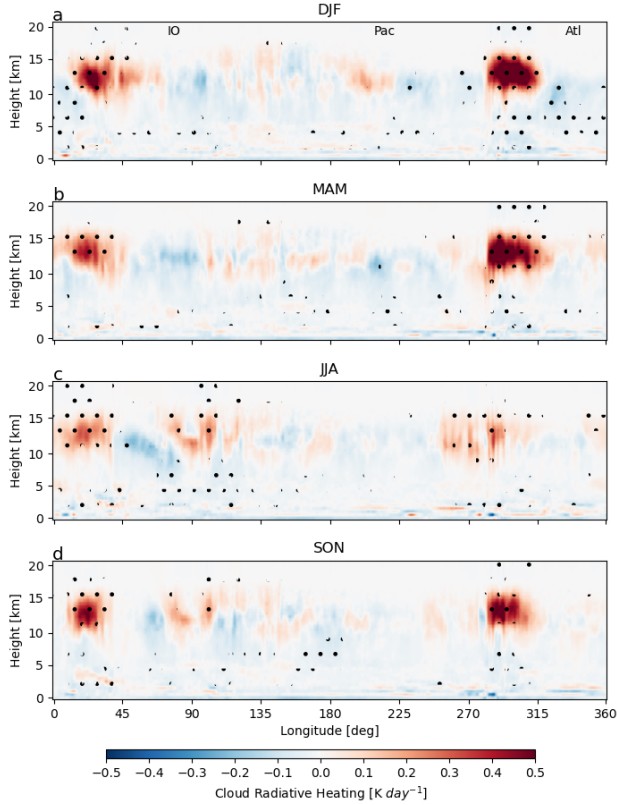

**Figure 3.** Cloud radiative heating difference between the PRIMAVERA version of EC-Earth with high and standard-resolution (EC-Earth3P-HR minus EC-Earth3P). The differences where a student t-test has a $95\%$ confidence based on monthly mean values are marked with black dots.

and the Asian monsoon in JJA, are present in the EC-Earth3P-HR results. In the lower troposphere, the net cooling at the top of the marine stratocumuli, more pronounced in JJA and SON, is also captured by the model.

There are, however, noticeable differences between the satellite retrievals and the model simulations. First, within most parts between $10$ and $15km$, the magnitude of the CRH is much stronger in EC-Earth3P-HR, almost twice as high as in the satellite observations. The seasonally averaged heating from the model-simulated convection sometimes reaches above $2Kday^{-1}$ while

in the satellite data, it typically stays around $1Kday^{-1}$. This overestimate by the model occurs over all convectively active regions, but the CRH is also overestimated in the stratocumulus regions, where the modelled cooling is too weak. The second striking disagreement between the model and observations relates to the vertical structure. The modelled CRH is vertically located in the upper troposphere, predominantly $12$–$14km$, where convective outflow and detrainment typically occurs. In the middle troposphere the CRH from the model simulations is weaker than in the observations. It should be noted that these

disagreements are visible even though the satellite observations are matched to the interpolated CRHs from the nearest 3-hourly time steps in the model simulations.





Figure 3 address the question if the high-resolution version of EC-Earth offer any improvement when compared to its standard counterpart, showing the difference in CRH between EC-Earth3P-HR and EC-Earth3P. Over the American and the African continents, there are considerable differences for the simulations with a lower resolution. When the convection is
strong, the CRH in EC-Earth3P-HR is $0.5 K day^{-1}$ higher compared to EC-Earth3P. During the dry period (JJA), the differences are reduced, but still reaching up to $0.3 K day^{-1}$. Similarly, when the Indian monsoon is at its peak, during JJA, the CRH is larger in EC-Earth3P-HR than in EC-Earth3P. Even though the convection over the west Pacific and the East Indian Ocean is persistently active, there are only small differences between the two model versions over this area.

The difference in CRH between EC-Earth3 and EC-Earth3P is shown in Figure 4. EC-Earth3 has a slightly higher CRH over
the American continent for all seasons ($0.3$–$0.4 K day^{-1}$) and for the west Pacific Ocean in MAM ($0.2 K day^{-1}$). On the other hand, EC-Earth3P has slightly higher CRH over the Indian Ocean, especially during SON. The maritime stratocumulus regions have slightly higher CRH in EC-Earth3P than in EC-Earth3. The differences are in general small, with low significance, but both EC-Earth3 and EC-Earth3P are closer to the satellite observations than EC-Earth3P-HR, despite the higher horizontal resolution in the latter.

### 3.1.2 Meridional and Zonal average

In order to understand the differences between model simulations and observations, we explore the vertical structure of CRH and the clouds (Figures 5 and 6). The plots show the CRH divided into its shortwave and longwave components, as well as the vertical structure of cloud fraction and cloud water content, for all four seasons.

Examining the vertical structure of the shortwave and longwave components of CRH reveals clear differences (Figure 5). In
the lowermost troposphere (up to about $2 km$), all model versions overestimate shortwave CRH ($0.2 K day^{-1}$ difference), while the CRH is strongly underestimated in the middle and upper troposphere (up to $12 km$) by $\sim -0.4 K day^{-1}$. Above $12 km$, the differences are negligible. These shortwave over- and underestimates are to some extent compensated in the longwave spectrum by overestimated cooling in the lowermost troposphere and heating in the middle-to-upper troposphere. The resulting net CRH shows stronger heating for the satellite $\sim 0.4 K day^{-1}$) between 2 and $10 km$ compared to the models ($\sim 0.2 K day^{-1}$) while
the net CRH in the upper troposphere (between $10$–$14 km$) is higher in the models ($1 K day^{-1}$) compared to the observations ($0.5 K day^{-1}$). This large difference in the upper troposphere is mainly due to the strong overestimate of longwave heating by the models and appears in all seasons.

Figure 6 shows that the models underestimate the cloud fraction as well as cloud liquid and ice water content in the lower and middle parts of the troposphere. This underestimate results in suppressed shortwave heating and longwave cooling in the
middle troposphere in the models, thus, at least partly, explaining the differences observed Figure 5. It is interesting to note that even though the difference in cloudiness is substantial in the middle part of the troposphere between the models ($0.05$) compared to satellite ($0.1$), the difference in liquid water content is smaller which implies that the models have fewer but denser clouds compared to observations. In the upper troposphere, although the models do have similar cloudiness as in the observations, the ice water content is much lower in the models (peaks at $0.002 g m^{-3}$) compared to observations ($0.011 g m^{-3}$).
This leads to a significant underestimate of cloud top longwave cooling in the models in the upper troposphere (Figure 5). The



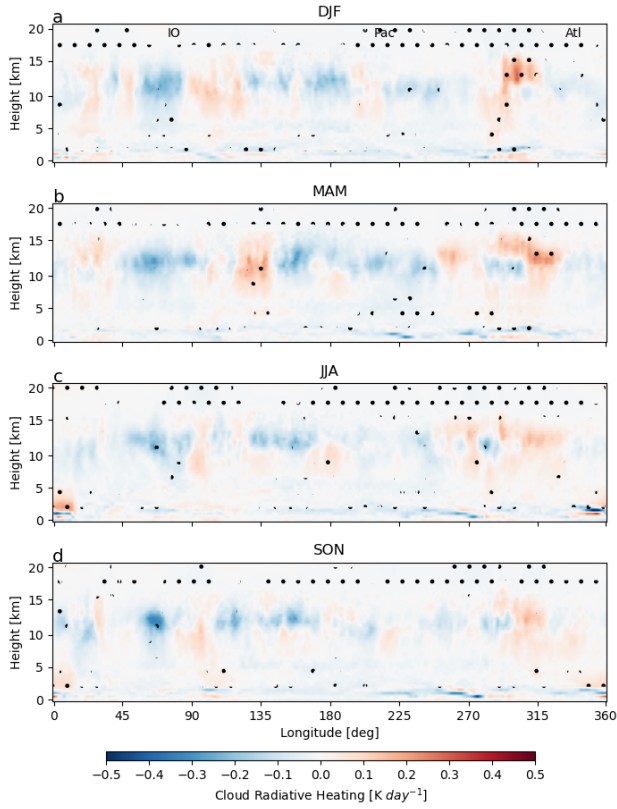

**Figure 4.** Cloud radiative heating difference between EC-Earth model version 3.3.1 and the PRIMAVERA standard-resolution (EC-Earth3 minus EC-Earth3P). The differences where a student t-test has a $95\%$ confidence based on monthly mean values are marked with a black dot.

underestimate of cloudiness seen in EC-Earth and the corresponding underestimate of shortwave heating are in line with the results reported from previous studies with different climate models or satellite data (e.g. Wang and Su, 2013; Cesana et al., 2019).

## 3.2 Cloud radiative heating in response to ENSO phase

### 3.2.1 Meridional average

Here, the CRH response to the positive and negative phases of ENSO (ENSOP and ENSON) is investigated. The CRH anomalies relative to the mean of the investigated period are averaged between latitude $\pm15°$ and are displayed in Figures 7 to 10, similar to Figures 1 to 4.

The observed ENSO-driven CRH anomalies from the satellite observations are shown in Figure 7. The meridional changes in deep convection in response to the shifts in the Walker circulation during ENSO phases are clearly visible in the CRH



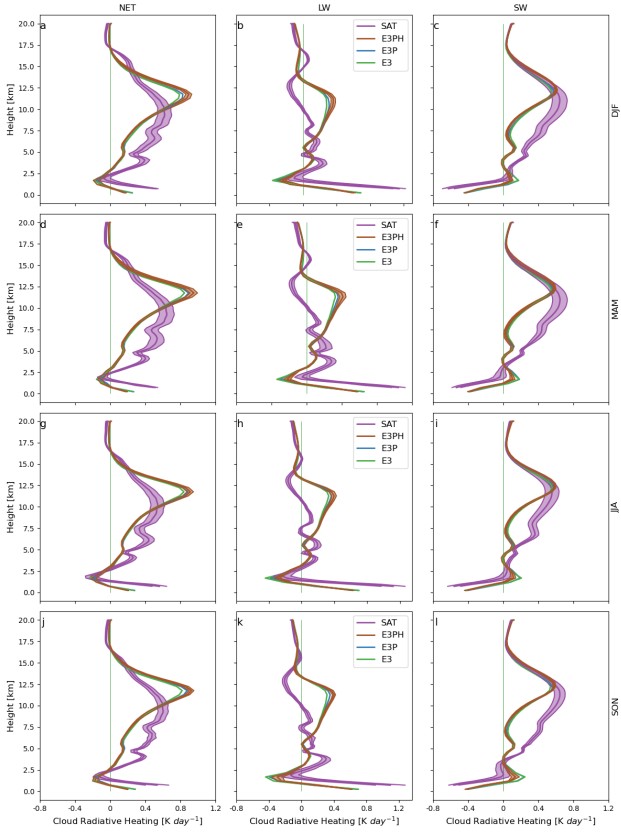

**Figure 5.** Vertical cloud radiative heating for EC-Earth3P-HR (E3PH), EC-Earth3P (E3P), EC-Earth3 (E3) and satellite observations (SAT) averaged over the entire tropics for each height bin. Column 1: net (SW+LW; NET) heating, column 2: longwave (LW) heating, and column 3: shortwave (SW) heating, divided into four seasons. Shaded area show $\pm 1\sigma$ spread for satellite and EC-Earth3P-HR, based on monthly mean values.

anomalies. During ENSOP, trade winds over the Pacific weaken, allowing a surge of warm water eastwards, leading to colder sea surface temperatures (SST) in the western Pacific Ocean while central and eastern Pacific will experience warmer than usual SST (see Timmermann et al., 2018). The increase in SST leads to additional and more vigorous convection. These will, in turn, generate stronger atmospheric radiative heating ($0.75 K day^{-1}$) in the central Pacific atmosphere. Over the western Pacific, the decrease in SST will instead generate less convection, leading to a decrease in the CRH ($-0.5 K day^{-1}$). In contrast, for ENSON, the surface water in the western Pacific and eastern Indian Ocean is warmer than usual, while central and eastern parts of the Pacific Ocean are colder than usual. This leads to an increase of CRH with $0.25 K day^{-1}$ over West Pacific and East Indian Ocean and a decrease with $-0.25 K day^{-1}$ over the central Pacific Ocean. The response to ENSON is expected to be smaller than the response to ENSOP when compared to the average since ENSON is an enhancement of the normal mode.




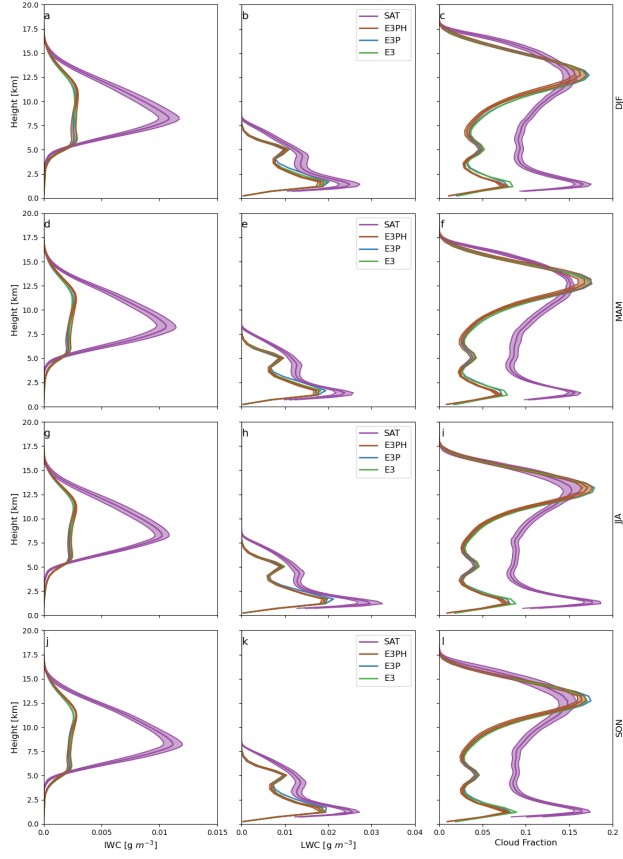

**Figure 6.** Cloud water content and cloud fraction in EC-Earth3P-HR (E3PH), EC-Earth3P (E3P), EC-Earth3 (E3) and Satellite observations (SAT) averaged over the entire tropics for each height bin. Column 1: ice water content (IWC), column 2: liquid water content, and column 3: cloud fraction (CF), divided into four seasons. Shaded area show $\pm 1\sigma$ spread for satellite and EC-Earth3P-HR, based on monthly mean values.

ENSO also has implications for clouds over the Atlantic Ocean. Madenach et al. (2019) found that close to the equator, high clouds in the Atlantic decreased during ENSOP and increased during ENSON, while the opposite was observed for low-level clouds. For ENSOP the CRH decreases by $-0.5 K day^{-1}$ in the Atlantic middle and high troposphere and the radiative cooling due to the maritime stratocumulus clouds outside Angola increases with $-0.75 K day^{-1}$.

    Figures 8a–b shows the vertical structure of CRH for the EC-Earth3P-HR simulation and Figures 8c–d shows the anomaly
differences between EC-Earth3P-HR and satellite observations, similar to Figure 7. While the shift in the Walker circulation and the sign of the CRH anomalies are captured in the EC-Earth3P-HR simulation, there are again some striking differences compared to the satellite data. The magnitudes of cloud radiative heating and cooling rates are almost two times larger in the EC-Earth3P-HR simulation in the active regions of ENSO. The CRH anomalies in the model are mostly restricted to the upper troposphere, whereas in the satellite observations, the entire troposphere experiences changes in the CRH. The maritime



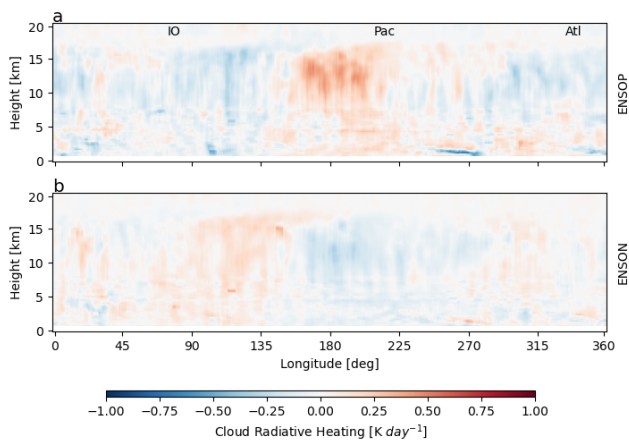

**Figure 7.** Cloud radiative heating from satellite during the positive (a) and negative (b) phases of ENSO compared to the average between latitude $\pm 15°$. There is a simple moving average filter applied to the plots to reduce noise.

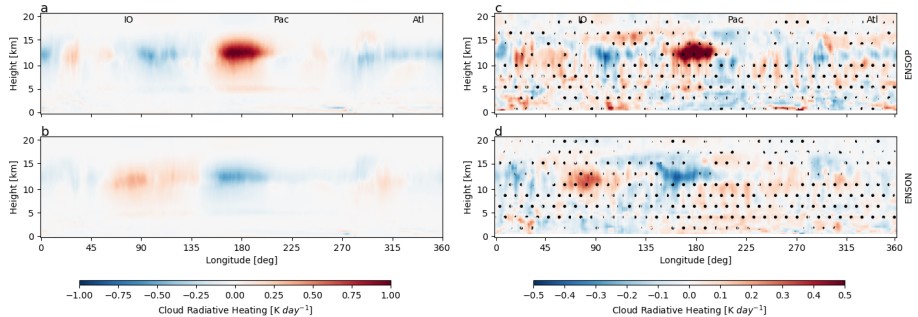

**Figure 8.** As in Figure 7, but showing the cloud radiative heating anomalies from EC-Earth3P-HR during positive (a) and negative (b) ENSO compared to the average, and cloud radiative heating anomaly differences between satellite observations and EC-Earth3P-HR (EC-Earth3P-HR minus satellite ) during positive (c) and negative (d) ENSO. The differences where a student t-test has a $95\%$ confidence based on monthly mean values are marked with a black dot (c-d).

stratocumulus regions off the coasts of south-central America show similar behaviour as in the satellite data, while the changes in CRH west of Africa are absent in the model. Over the Atlantic Ocean, there is a strong cooling ($-0.75 K day^{-1}$) above $10 km$ during ENSOP, while the result for ENSON is close to neutral.

    The differences in CRH anomalies between EC-Earth3P-HR and EC-Earth3P, and EC-Earth3 and EC-Earth3P are shown in Figures 9 and 10. Similar to the result in Figure 3, the magnitudes of radiative heating/cooling rate anomalies are higher in the

EC-Earth3P-HR simulation compared to EC-Earth3P, for both the ENSOP and ENSON phases. Central Pacific, Africa, and South America all have a stronger CRH anomaly ($0.5 K day^{-1}$ difference) in EC-Earth3P-HR during ENSOP. During ENSON

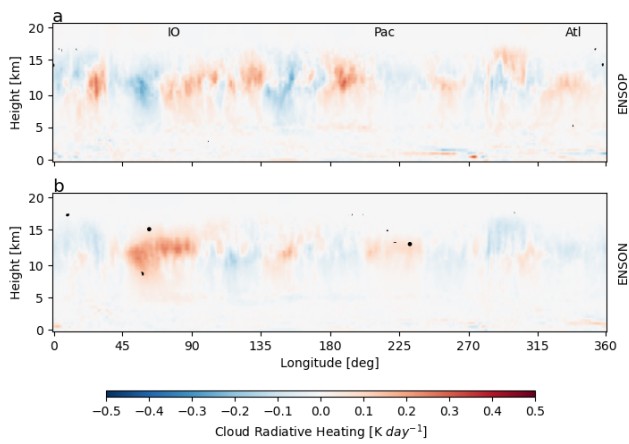

**Figure 9.** Cloud radiative heating anomaly differences between the PRIMAVERA versions of EC-Earth with high and standard-resolution (EC-Earth3P-HR minus EC-Earth3P) during positive (a) and negative (b) ENSO. The differences where a student t-test has a 95% confidence based on monthly mean values are marked with a black dot.

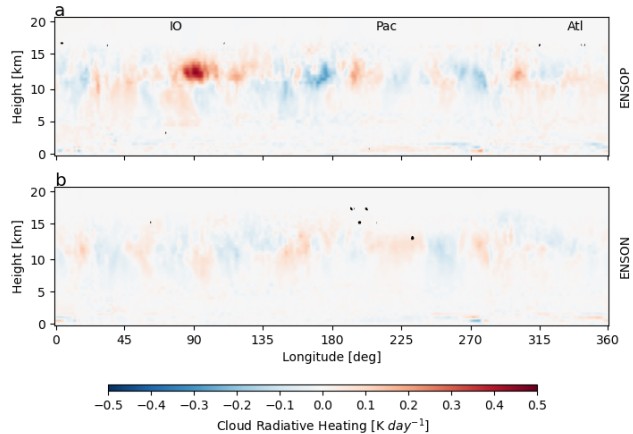

**Figure 10.** Cloud radiative heating anomaly differences between EC-Earth model version 3.3.1 and the PRIMAVERA version with standard-resolution (EC-Earth3 minus EC-Earth3P) during positive (a) and negative (b) ENSO. The differences where a student t-test has a 95% confidence based on monthly mean values are marked with black dots.

there is a larger CRH over the Indian Ocean in EC-Earth3P-HR while the differences are small in the Pacific Ocean and the Atlantic.

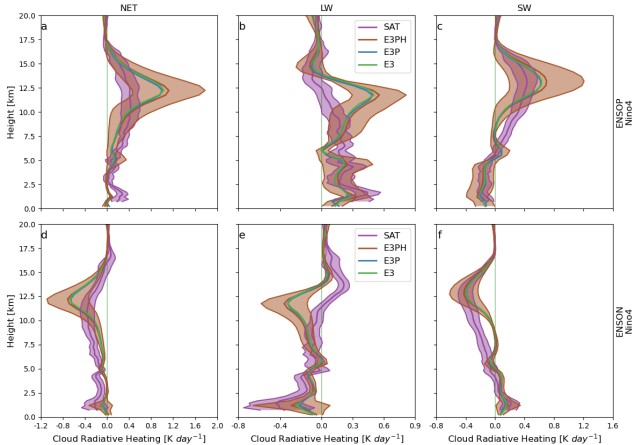

**Figure 11.** Vertical cloud radiative heating anomalies for EC-Earth3P-HR (E3PH), EC-Earth3P (E3P), EC-Earth3 (E3) and satellite observations (SAT) averaged over the Nino4 region for both ENSOP and ENSON. Column 1: net (SW+LW; NET) heating, column 2: longwave (LW) heating, and column 3: shortwave (SW) heating. Shaded area show $\pm 1\sigma$ spread for satellite and EC-Earth3P-HR, based on monthly mean values.

The CMIP6 version of EC-Earth (EC-Earth3) shows higher CRH anomalies in the western Indian Ocean ($0.75 K day^{-1}$)
compared to EC-Earth3P and values are even larger compared to the high-resolution version EC-Earth3P-HR (Figure 9).
Over the Pacific, the differences between EC-Earth3 and EC-Earth3P shows the same tendency as the difference between
EC-Earth3P-HR and EC-Earth3P. Away from the ENSO region, there is no clear difference in EC-Earth3 compared to EC-
Earth3P. In the case of ENSON, the differences between these two versions of the model remain minimal. Overall, differences in
anomalies between EC-Earth3 and EC-Earth3P are small and with no significance according to the student t-test. It is important
to be aware of that for a $95\%$ confidential interval, $5\%$ of the pixels will be miss-classified. This random miss-classification is
probably responsible for the few black dots in Figure 9 and 10.

### 3.2.2 Nino 3.4 region

The Nino 3.4 region, subregion to the two areas, Nino3 ($5°N$–$5°S$, $150°W$–$90°W$) and Nino4 ($5°N$–$5°S$, $160°E$–$150°W$),
is used to determine if there is an El Niño or La Niña event. Both the Nino3 and Nino4 areas experience large SST variability
during ENSOP and ENSON periods and are therefore useful to investigate in more detail. This will further help to interpret the
differences observed in Figures 8–10. The vertical structure of CRH anomalies during both ENSOP and ENSON, separated
into its shortwave and longwave components, are plotted in Figure 11 for Nino4 and Figure 12 for Nino3, while the anomalies
for the cloud properties are plotted in Figures 13 and 14, similar to Figures 5 and 6.

During ENSOP, the models show a strong increase in shortwave heating peaking at $12 km$ in the Nino4 area. Even though the
peak CRHs are in general similar among the models, the EC-Earth3P-HR shows the largest anomalies, reaching $0.8 K day^{-1}$,
while EC-Earth3 and EC-Earth3P have anomalous heating rates in the order of $0.7 K day^{-1}$. This anomalous heating in the

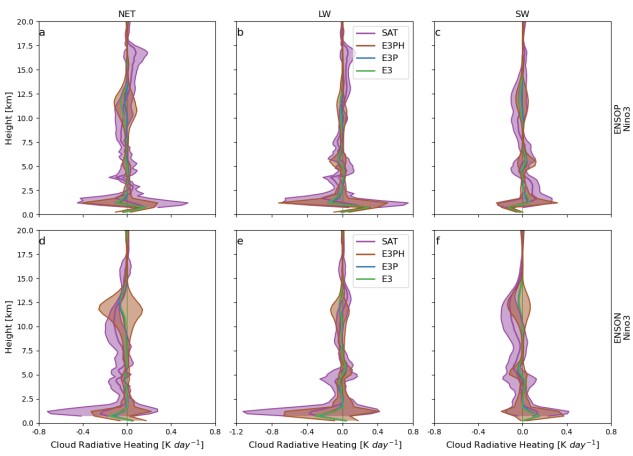

**Figure 12.** Same as Figure 11 but averaged over the Nino3 region.

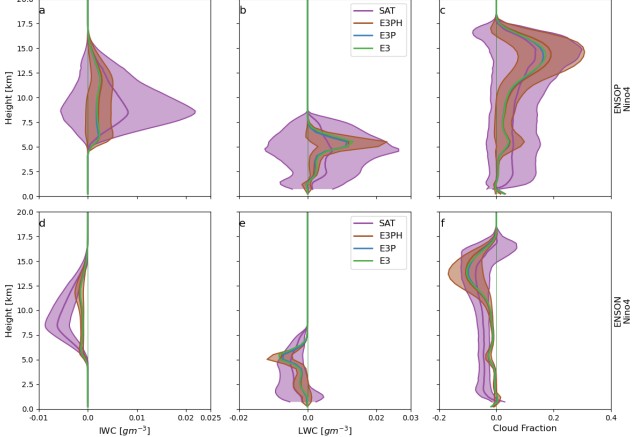

**Figure 13.** Cloud water content and cloud fraction for EC-Earth3P-HR (E3PH), EC-Earth3P (E3P), EC-Earth3 (E3) and satellite observations (SAT) averaged over the Nino4 region for both ENSOP and ENSON. Column 1: ice water content (IWC), column 2: liquid water content (LWC), and column 3: cloud fraction (CF). Shaded area show $\pm 1\sigma$ spread for satellite and EC-Earth3P-HR, based on monthly mean values.

models is up to twice as large as for the satellite data, showing a peak in shortwave anomalous heating at $0.4 K\,day^{-1}$. Furthermore, in the models, the peak in the shortwave anomaly is restricted vertically to the uppermost troposphere, while in the satellite data, increased shortwave heating during ENSOP is visible both in the middle and upper troposphere. This result can
be explained by the fact that the vertical cloud distribution in the models is predominantly restricted vertically to the uppermost troposphere, while in the observations, large cloud variability is also observed in the middle and lower troposphere (Figure 13).

The most considerable disagreement between the satellite observations and the model versions is seen in the longwave component of CRH in the upper troposphere. Here, the satellite data show a clear decrease in longwave heating with increasing altitude, mainly due to strong longwave cooling at the tops of the excessive convective and stratiform clouds and the optically



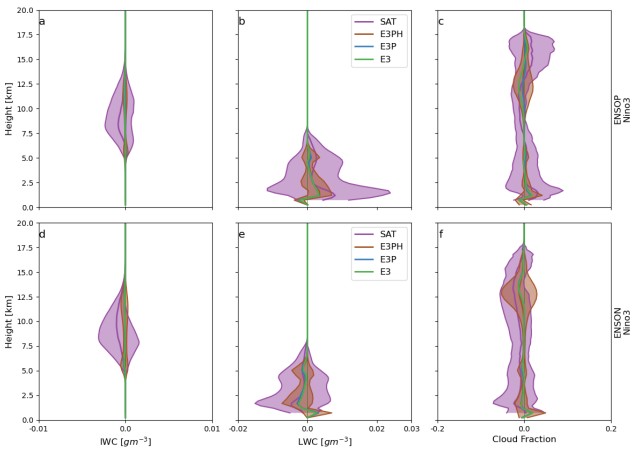

**Figure 14.** Same as Figure 13 but averaged over the Nino3 region.

thick cirrus. In the model simulations, the CRH anomaly continues to increase with height and then changes sign abruptly compared to the observations at around $14km$. As a result of these overestimations of both shortwave and longwave heating, the net CRH anomaly in the models ($1.2Kday^{-1}$) is almost three times higher compared to observations ($0.4Kday^{-1}$) in the upper troposphere in ENSOP for the Nino4 region.

As the Walker circulation is enhanced during the ENSON phase, the reduction in convection and associated clouds lead to a reduction in CRH in the upper troposphere over the Nino4 region (Figures 11d–f). In this case, although the reduced shortwave heating (peak at $-0.4Kday^{-1}$) is captured relatively well by the models, the magnitude of the reduction in longwave heating ($-0.4Kday^{-1}$), in the models, is large enough to result in a net CRH anomaly that is half of that from the observations ($-0.8Kday^{-1}$ compared to $-0.4Kday^{-1}$). Over the Nino3 region the CRH and cloud property anomalies are low both in the observations and model simulations, during both phases of the ENSO (Figure 12).

Overall the models tend to underestimate the anomaly in cloud ice water content and overestimate the change in liquid water content for both phases of ENSO over both the Nino3 and Nino4 areas (Figures 13 and 14). The variability in the satellite observations is however large for both liquid and ice water content compared to all three versions of the model investigated here.

## 4   Conclusions

In the past, cloud radiative forcing in climate models has mainly been evaluated at the top of the atmosphere or the surface since the net fluxes are more readily observed there, either by satellite or by surface instruments. A rigorous evaluation of the vertical structure of cloud radiative heating/cooling has only been possible since the last few years as the retrievals from the combined active radar and lidar sensors onboard CloudSat and CALIPSO satellites have matured enough to allow a quantitative analysis (Cesana et al, 2019). Understanding the vertical structure of CRH and evaluating climate models from this aspect is especially





important in the tropics. In this region, the CRH does not only impact the zonal and meridional temperature gradients, and thereby influencing the atmospheric circulation, but it also influences the troposphere-to-stratosphere transport.

In this study, we therefore investigated two versions of the EC-Earth climate model, where one version was used with two different horizontal resolutions (EC-Earth3P-HR and EC-Earth3P). In addition to a traditional statistical comparison, we also carried out a more process-oriented evaluation by examining how different versions of the EC-Earth model simulate CRH

during different phases of ENSO. The following conclusions can be drawn from the evaluations:

(a) Compared to the satellite data, all versions of EC-Earth capture the general structure of the seasonal and spatial variability in net CRH over both the convectively active regions and the maritime stratocumulus regions.

(b) The shortwave component of the radiative heating is overestimated by all model versions in the lowermost troposphere and underestimated in the middle troposphere. In the upper troposphere, the differences in shortwave heating are negli-

gible. These over- and underestimates of shortwave heating are partly compensated in the models by an overestimate of longwave cooling in the lowermost troposphere and heating in the middle and upper troposphere.

(c) There are two noticeable differences between EC-Earth (EC-Earth3P-HR) and the satellite retrievals. First, the magnitude of net CRH (above $2 K day^{-1}$) is much stronger in the model in the upper part of the troposphere over convectively active zones, almost twice as high as in the satellite observations. Second, the net CRH in the models is vertically limited to the

upper troposphere, predominantly between $12$–$14 km$, whereas, the satellite observations also show pronounced heating in the middle troposphere. These disagreements are the largest in EC-Earth3P-HR.

(d) There are substantial differences in the vertical structure of cloud fraction and cloud water content between the models and observations. In the upper part of the troposphere, the cloud fraction is similar but the models have less ice water content. In the lower and middle parts of the troposphere, the models underestimate the cloud fraction. These differences

are vital in understanding the disagreements in the magnitude and vertical distribution of the CRH between the models and the observations.

(e) The spatial CRH variability associated with the ENSO phases and shifts in the Walker circulation is also reasonably captured by the models. However, the differences in the magnitude and vertical structure of the CRH between the models and satellite data mentioned above remain or even increase.

Our results highlight the importance of having satellite observations that resolve the vertical structure of clouds for evaluating climate models and the importance of realistically simulating the vertical structure of cloud properties. We could, however, only notice negligible differences in the simulation of vertical cloudiness and CRH over most parts of the tropics when changing the horizontal resolution in EC-Earth. An increase in the vertical resolution could potentially further improve the representation of clouds and CRH in the models and such an investigation would be of interest in a prospective study. A longer time series from

the observations, especially for the ENSO analysis, would be preferable for future studies.



*Code and data availability.* At the time for the writing the CloudSat data could be freely downloaded from the website http://www.cloudsat.cira.colostate.edu. Access to the source code of the EC-Earth3 model is restricted and can be granted after registration. For further information contact the EC-Earth community at http://www.ec-earth.org. The code for analysing the satellite and model data can be found at https://github.com/smhi-erik/compare_sat_model (last access: 2020-06-11). Information regarding model configurations, data

availability or the code for analysing the data are available from the authors upon request.

*Author contributions.* The first author ran the model simulations and did the subsequent analysis and drafted the manuscript. Abhay Devasthale, Michael Tjernström and Annica M. L. Ekman were all part of designing the study. Klaus Wyser provided valuable input for running the model and added a climate modeller's perspective while Tristan L'Ecuyer provided insight to the CloudSat data. All the co-authors contributed to the interpretation of the results and reviewing the manuscript.

*Competing interests.* The authors declare that they have no conflict of interest.

*Acknowledgements.* Erik Johansson and Abhay Devasthale would like to acknowledge funding from the Swedish National Space Board (SNSB; contract Dnr: 84/11:1 and 84/11:2). The EC-Earth simulations for this publication were enabled by resources provided by the Swedish National Infrastructure for Computing (SNIC) at NSC. We are grateful to the CloudSat Data Processing Center (DPC) and Science Teams for providing CloudSat datasets.





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
