# Peer review of "Vertical structure of cloud radiative heating in the tropics: Confronting the EC-Earth v3.3.1/3P model with satellite observations"

_Geoscientific Model Development, 2020_

## Short Comment (SC1) · 14 Nov 2020

Dear authors,

in my role as Executive editor of GMD, I would like to bring to your attention our Editorial version 1.2:

https://www.geosci-model-dev.net/12/2215/2019/

This highlights some requirements of papers published in GMD, which is also available on the GMD website in the 'Manuscript Types' section: http://www.geoscientific-model-development.net/submission/manuscript_types.html

In particular, please note that for your paper, the following requirements have not been met in the Discussions paper:

- "The main paper must give the model name and version number (or other unique identifier) in the title."

- Code must be published on a persistent public archive with a unique identifier for the exact model version described in the paper or uploaded to the supplement, unless this is impossible for reasons beyond the control of authors. All papers must include a section, at the end of the paper, entitled "Code availability". Here, either instructions for obtaining the code, or the reasons why the code is not available should be clearly stated. It is preferred for the code to be uploaded as a supplement or to be made available at a data repository with an associated DOI (digital object identifier) for the exact model version described in the paper. Alternatively, for established models, there may be an existing means of accessing the code through a particular system. In this case, there must exist a means of permanently accessing the precise model version described in the paper. In some cases, authors may prefer to put models on their own website, or to act as a point of contact for obtaining the code. Given the impermanence of websites and email addresses, this is not encouraged, and authors should consider improving the availability with a more permanent arrangement. Making code available through personal websites or via email contact to the authors is not sufficient. After the paper is accepted the model archive should be updated to include a link to the GMD paper.

Therefore, please add the version number of EC-Earth model to the title.

Additionally, as GitHub is not a persistent archive, please provide a persistent release for the exact source code version used for the publication in this paper. As explained in https://www.geoscientific-model-development.net/about/manuscript_types.html the

preferred reference to this release is through the use of a DOI which then can be cited in the paper. For projects in GitHub a DOI for a released code version can easily be created using Zenodo, see https://guides.github.com/activities/citable-code/ for details. Finally note, that according to our new Editorial (v1.2) all data and analysis / plotting scripts should be made available.

Yours,

Astrid Kerkweg

---

## Author Comment (AC1) · 30 Nov 2020

Dear Astrid Kerkweg,

Thank you for making us aware of the latest guidelines. Since we can not make changes to the original submitted manuscript, we will during the review process add the model version number to the title, e.g. "Vertical cloud radiative heating in the tropics: Confronting the EC-Earth v3.3.1/3P model with satellite observations".

As the review process comes to an end, we will upload our model data and final analysis scripts to Zenodo and update the section "Code and data availability" with the DOI

numbers.

Best Regards Erik Johansson on behalf of all co-authors.
* * *

---

## Referee Comment (RC1) · Anonymous Referee #1 · 5 Dec 2020

**General comments**

This paper represents a comparison of cloud radiative heating rates and cloud vertical profiles in the tropics between the EC-Earth model and datasets derived from active satellite measurements. Differences between three configurations of EC-Earth (two model versions, one of them run at two resolutions) are also compared.

These kind of comparisons are still relatively rare, especially for the radiative heating rates, so in principle this study is a useful contribution. However, having read the paper carefully, I feel somewhat dissatisfied about it. The paper documents many results,

but with rather little of deeper-going analysis. Perhaps the paper would benefit from a sharper focus on a comparison of a single version of EC-Earth with the observations. The differences between the three configurations of EC-Earth are small compared to the model-vs-observation differences, and for the most part, not even statistically significant for these relatively short runs. At least I would restrict the ENSO-related analysis to a single model configuration. Another concern — and indeed a source of irritation — is that the readability of many of the figures is quite poor (see the technical comments). The English language is generally good, however.

**Specific comments**

1. line 12: I suppose this refers to the upper troposphere?

2. line 56: "poor spatial resolution". Do you mean "poor spatial coverage"?

3. lines 61–64: The introduction does not put the present paper properly in the context of previous research. In particular, the paper by Cesana et al. (2019) entitled "The vertical structure of radiative heating rates: a multimodel evaluation using A-train satellite observations" should be discussed here briefly (this paper is now mentioned only in the Conclusions). What is the novelty / additional value in the current paper compared to Cesana et al.? Analysis related to ENSO, perhaps?

3. lines 67–77: The most relevant point for the present paper is what is the difference between the PRIMAVERA version of EC-Earth and EC-Earth v. 3.3.1. Perhaps this is said on lines 76–77, but it should be formulated more clearly.

4. lines 81–82: "The vertical levels are not equally distributed throughout the atmosphere." This sentence is not necessary, since all atmospheric GCMs have non-uniform

vertical grids.

5. lines 115–117: "CloudSat and CALIPSO pass the equator at roughly 13:30 local time during daytime, so the model results are linearly interpolated from the two nearest output times to the fit the satellite overpass time". Is it indeed so, that your results only represent local daytime (close to 13:30)? (The relatively large SW CRH values compared to LW CRH in Fig. 5 suggest to me that this might be the case.) This would have a major effect on the SW CRH, as the near-noon SW CRH is much larger than the diurnal-mean CRH, and also severely bias the net CRH. A procedure to mitigate this bias, that is, to calculate approximate diurnal-mean results, was introduced in your earlier work (Johansson et al. 2015, cited in the manuscript). Why not to use it in the present work? See also the discussion on p. 1575 in Cesana et al. (2019).

6. line 121: I assume that the clouds in the lowest 750 m are included in the computation of model heating rates. Please mention this explicitly.

7. Figures 1-2: These figures are difficult to interpret because the net CRH consists of SW and LW components, which may even partly oppose each other. I therefore recommend adding figures which show separately the SW, LW and net components of CRH. To avoid an excessive number of figure panels, it would be sufficient to show only the annual-mean results. (I am aware of Fig. 5, but since it shows mean values over the entire tropics, all regional features are lost).

8. A further suggestion would be to analyze the CRHs as a function of sea surface temperature, or mid-tropospheric vertical velocity, to distinguish between convective and non-convective regions (see also the analysis in Cesana et al. 2019). It is up to the authors to decide if (and how) they wish to pursue this suggestion.

9. line 168: "peaks in cloud fraction in August – October". Does this sentence refer

to marine stratocumulus in general, or the southern hemisphere (Peruvian, Namibian) stratocumulus regions only?

10. line 199: "...despite the higher resolution in the latter". This is actually not so surprising. I don't see any obvious reason why increasing resolution would automatically lead to substantial changes, nor why these changes would show up as improvements in large-scale features when compared to observations (it mainly depends on your luck!). The situation might be different if the resolution was high enough to explicitly resolve deep convection, but even the finer resolution considered here (40 km) is far too coarse for that.

11. lines 211, 219: The overly positive LW CRH in the upper troposphere is a curious feature. It is suggested that this is due to underestimated ice water content, which leads to underestimated cloud top LW cooling. This does not however explain why the modelled LW CRH is positive (middle column of Fig. 5). One possibility is that due to underestimated cloud fraction in the midtroposphere, too much upwelling LW radiation from lower levels reaches the high clouds, leading to LW heating.

12. line 219: Regarding the large underestimate of IWC in EC-Earth compared to the satellite observations, I am wondering if precipitating ice is included in the IWC in the latter. In models it is generally not. There are also other potential reasons that could make the satellite profiles and the model profiles in Fig. 6 not to be fully compatible. For example, there is no satellite simulator in the model. Also, do the EC-Earth cloud fraction and cloud water fields include convective clouds?

13. In Figures 5 and 6, the results seem much the same for all seasons. Furthermore, in the text on lines 201–223, the seasonal differences are not discused at all. So why not simply show the annual-mean values? This would also allow combining Figs. 5 and 6 into one figure with six panels.

14. line 247: this should be "radiative heating and cooling rate anomalies"?

15. line 251: "a strong cooling (-0.75 $\mathrm{K\,day^{-1}}$) above 10 km during ENSOP". I cannot find this large CRH anomaly over the Atlantic in Fig. 8, and at least it is not representative of the Atlantic region in general. Please also check that the other numerical values given in the text in Section 3.2.1 are consistent with Figs. 7 and 8.

16. lines 253–266 and Figs. 9 and 10. The comparison of ENSO-related cloud radiative heating anomalies between different versions of EC-Earth seems largely useless, owing to the low statistical significance of the results. The low significance itself is not surprising considering your small sample size. I recommend to eliminate this part of the manuscript.

17. line 267: The section title "Nino 3.4 region" is misleading, when you discuss separately Nino 3 and Nino 4 but not Nino 3.4.

18. caption of Fig. 13: This should be "Cloud water content and cloud fraction anomalies".

19. line 292: "half of that" should be "twice that"?

20. line 304: It would be relevant to comment on how your findings compare with those seen in the multimodel study of Cesana et al. (2019).

21. line 317: Do you mean "two noticeable differences in net CRH"?
**Technical and language corrections**

1. Figures 2, 5, 6, 8, 11, 12, 13, and 14, and especially their labels, are painfully small to read. Please enlarge them. Too small figures are a sure method to make a reader (and a reviewer) *förbannad*.

2. The choice of colours in Figs. 5, 6, 11, 12, 13, and 14 is not good. It requires effort to distinguish SAT (a colder shade of purple) from E3PH (a warmer shade of purple/red). Please use clearly distinguishable colours.

3. line 284: Replace "excessive" with "extensive". "Excessive" implies that there is too much cloudiness in the observations.

---

## Referee Comment (RC2) · Anonymous Referee #2 · 10 Feb 2021

This study evaluates the performance of various versions of the EC-Earth model against satellite retrievals from CloudSat/Calipso focusing on the clouds and cloud radiative heating rates (CRH). The authors evaluate the intraseasonal and interannual variability of CRH in the tropics. This study is useful and would be a valuable contrition to the related and increasing body of literature that deals with the coupling between the circulation and CRH. However, the authors mostly illustrate a large amount of results without providing enough insights and deeper analysis of the results. Additionally, my impression was that the authors did not properly illustrate the gained knowledge from this study (potentially by showing many results without a clear story?) and did not communicate the main key points of this study to the reader. Another general remark could

be the fact that the comparison among the different model versions was not giving any extra information to the paper and thus the authors should reconsider to focus on a specific model version and try to provide more deeper analysis and explanations for the differences between the EC-Earth model and the observations. As a final major remark, the readability of the vertical profiles was very bad and the authors should reconsider using more distinguishable colors.

Major comments:

1) There is a confusing part in this study that comes from the fact that the authors do not clearly explain the method that they use to compare the models with the observations. Is it monthly mean, 3-hourly data that the authors use to compare the simulated CRH with the observed ones?

2) Another general remark is that the authors in the results section mostly describe the figures without conceptually connecting the key points of the study and without providing more insights and explanations.

3) The authors illustrate the changes in CRH across seasons and different ENSO phases, however they do not illustrate the gained knowledge from this changes and do not properly communicate the key message from these comparisons in the context of previous studies.

4) The readability of the figures is really bad (particularly those that include vertical profiles), thus more distinguishable colors should be used.

Minor comments:

Line 3: Typo: coupling

Line 148: Typo: analysis

Line 159: It could be easier for the reader if the figures and the text had different longitude values to something like 45E, 90E, 90W, 45W.

Figure 2 (e-h): To make the comparison easier since the differences are larger than 1K/day wouldn't it make more sense to use the same colorbar as in panels (a-d) so that the reader can compare the differences compared to the absolute values?

Line 179: Do the authors refer to the temperature tendency due to convective parameterizations or the CRH? It should be properly phrased here.

Line 180-181: Since the model overestimates the magnitude of the CRH over both convectively active and stratocumulus regions is there a reason to separate them by using "but"?

Lines 183-184: Any potential explanation for that?

Lines 184-186: This is confusing. Isn't it the case that panels (e-h) in Figure 2 show monthly mean differences between the model and the observations? See major comment 1

Lines 189-190: It is not obvious that the convection is strong. This is not shown anywhere. Moreover, the authors do not provide enough evidence when is the dry and the wet season for a location. Maybe show this somewhere?

Lines 187-199: Any further insights about these differences?

Lines 204-205: Isn't it the case that the magnitude in the lower troposphere (below 2 km) is smaller across the model versions, thus underestimating the magnitude of the cooling?

Lines 210-212: Any insights about these differences? Could you potentially explain why the models simulate the maximum heating higher that in the observations? Moreover, the different model versions seems not to be sensitive to CRH. Is there any explanation why these differences are so small?

Figure 6: It might be better to use letters to show which panels show IWC, LWC and CF.

Lines 218-220: This is somehow confusing. Despite CF differences in the upper troposphere are small, the models seems to underestimate IWC compared to the observations and and still producing stronger CRH than the observations. Could you explain why is this the case?

Lines 214-215: This underestimation of CRH is mainly evident in the mid-troposphere. However, in the lower troposphere CRH agree more despite the fact that cloud properties differ substantially. Any explanation regarding that?

Lines 244-253: Could the authors provide more insights that would explain these differences?

Figure 7: As before, either adjust the colorbar to make the comparison more straightforward or mention in the figure caption the different colorbar for the differences. Additionally, in the figure caption it would be more appropriate to mention the satellite (CloudSat/Calipso) instead of mentioning CRH from satellite.

Lines 259-260: Could the authors providence more evidence why is this happening?

Line 262: Why the differences are mostly evident over the ENSO region? Is this consistent with previous modeling studies? What could be the reason behind these differences?

Line 274: Which figure shows that? Maybe add the figure at the end of the phrase?

Line 282: As before, which figure shows that?

Lines 295-298: Maybe use the same wording for anomalies? Using anomalies and changes in the same phrase might be confusing for the reader.

---

## Author Comment (AC2) · 31 Mar 2021

**Response to Reviewer #1**:

We thank the reviewer for the constructive criticism that led to improvements in the manuscript. Following the suggestions from the both reviewers, we have carried out a major revision. The discussions and readability are improved and the redundancies in figures are removed. Among other things, Figs. 4, 5, 6, 8, and 11 are revised, while Figs. 9 and 10 are removed to declutter the discussion. Please note that few of the reviewer comments may not be valid anymore due to the revision of either the underlying figure or the text.

Please find below our point by point reply to your comments.

This paper represents a comparison of cloud radiative heating rates and cloud vertical profiles in the tropics between the EC-Earth model and datasets derived from active satellite measurements. Differences between three configurations of EC-Earth (two model versions, one of them run at two resolutions) are also compared.

These kind of comparisons are still relatively rare, especially for the radiative heating rates, so in principle this study is a useful contribution. However, having read the paper carefully, I feel somewhat dissatisfied about it. The paper documents many results, but with rather little of deeper-going analysis. Perhaps the paper would benefit from a sharper focus on a comparison of a single version of EC-Earth with the observations.

The differences between the three configurations of EC-Earth are small compared to the model-vs-observation differences, and for the most part, not even statistically signif-icant for these relatively short runs. At least I would restrict the ENSO-related analysis to a single model configuration. Another concern — and indeed a source of irritation — is that the readability of many of the figures is quite poor (see the technical comments). The English language is generally good, however.

We find it encouraging that the reviewer thinks that this is a useful contribution. The evaluation of radiative heating rates, although they are at the core of net radiation budget, is indeed rare and we would therefore like to highlight how well a typical climate model simulates them. We have revised the manuscript to include deeper analysis, to improve readability and the quality of figures.

In our analysis, we found that there are either no statistically significant differences in the heating rates between the high- and low-resolution EC-EARTH model versions or the biases are enhanced in the high resolution version. This is in fact one of the main points that we would like to highlight in the study that incorporating processes are more important than just improving the resolution. We do however understand how one could get a feeling of having a redundant information. Following the reviewer suggestion, we have presented the figures of ENSO-analysis only for one model version, while mentioning the results from the other version only in the text form in the revised manuscript.

**Specific comments**

1. line 12: I suppose this refers to the upper troposphere?
Yes, this is clarified in the revised manuscript.

2. line 56: "poor spatial resolution". Do you mean "poor spatial coverage"?

Yes. "Coverage" is a better word in this context.

3. lines 61–64: The introduction does not put the present paper properly in the context of previous research. In particular, the paper by Cesana et al. (2019) entitled "The vertical structure of radiative heating rates: a multimodel evaluation using A-train satellite observations" should be discussed here briefly (this paper is now mentioned only in the Conclusions). What is the novelty / additional value in the current paper compared to Cesana et al.? Analysis related to ENSO, perhaps?

We agree. We have now mentioned Cesana et al in the Introduction. The ENSO analysis presented here is entirely new and we also hope that our focus on the meridional differences is of interest to the scientific community. Furthermore, one of the model versions we compare is a part of the CMIP6 group and our new comparison is hopefully of interest to the climate modelling community. We also show the insensitivity of simulated heating rates to the model resolution, which is not shown in any study before.

3. lines 67–77: The most relevant point for the present paper is what is the difference between the PRIMAVERA version of EC-Earth and EC-Earth v. 3.3.1. Perhaps this is said on lines 76–77, but it should be formulated more clearly.

We have updated the paragraph to make it clearer what model versions we are using and the differences between them.

4. lines 81–82: "The vertical levels are not equally distributed throughout the atmosphere." This sentence is not necessary, since all atmospheric GCMs have non-uniform vertical grids.

It is true that for most people working with GCMs this fact is common knowledge. However, we still believe it can be important to point out this in this paper, since we have a strong focus on the vertical structure, to reduce the risk for misunderstandings.

5. lines 115–117: "CloudSat and CALIPSO pass the equator at roughly 13:30 local time during daytime, so the model results are linearly interpolated from the two nearest output times to the fit the satellite overpass time". Is it indeed so, that your results only represent local daytime (close to 13:30)? (The relatively large SW CRH values compared to LW CRH in Fig. 5 suggest to me that this might be the case.) This would have a major effect on the SW CRH, as the near-noon SW CRH is much larger than the diurnal-mean CRH, and also severely bias the net CRH. A procedure to mitigate this bias, that is, to calculate approximate diurnal-mean results, was introduced in your earlier work (Johansson et al. 2015, cited in the manuscript). Why not to use it in the present work? See also the discussion on p. 1575 in Cesana et al. (2019).

Yes, our results only represent local daytime (13:30). Cesna et al. used accumulated daily (and night separately) values from the models and therefore needed to normalise the satellite data in a similar way as we did in the Johansson et al. (2015). To compare daily mean values or instantaneous values (for a specific time) have its pros and cons and in this manuscript we decided to focus on the instantaneous values. Johansson et al. (2015) was based only on the observational data, but here the main aim is to evaluate a climate model. Given that the convective cloud regimes in the tropics have stronger diurnal cycles, we thought it would be better to do a comparison at the instantaneous level. This would then also highlight the differences arising from the out of phase diurnal cycles. This is clarified better in the revised manuscript.

6. line 121: I assume that the clouds in the lowest 750 m are included in the computation of model heating rates. Please mention this explicitly.

They are included in the computation, we only excluded them in our analysis. We have now clarified this.

7. Figures 1-2: These figures are difficult to interpret because the net CRH consists of SW and LW components, which may even partly oppose each other. I therefore recommend adding figures which show separately the SW, LW and net components of CRH. To avoid an excessive number of figure panels, it would be sufficient to show only the annual-mean results. (I am aware of Fig. 5, but since it shows mean values over the entire tropics, all regional features are lost).

This is a good idea and we have added these components separately in the revised version.

8. A further suggestion would be to analyze the CRHs as a function of sea surface temperature, or mid-tropospheric vertical velocity, to distinguish between convective and non-convective regions (see also the analysis in Cesana et al. 2019). It is up to the authors to decide if (and how) they wish to pursue this suggestion.

It is indeed an interesting idea and we thank the reviewer for pointing it out. We believe it deserves to be a separate exercise of its own. Given the limited time we have for revision, it would be difficult to fit it in the current context. If the Editor could grant us extra time to incorporate this analysis, we would be happy to do it.

9. line 168: "peaks in cloud fraction in August – October". Does this sentence refer marine stratocumulus in general, or the southern hemisphere (Peruvian, Namibian) stratocumulus regions only?

This statement refers to those stratocumulus regimes in the southern hemisphere. The seasonal peak for marine stratocumulus in northern hemisphere is usually a little bit earlier, May - June. We have clarified the sentence in the revised manuscript.

10. line 199: "...despite the higher resolution in the latter". This is actually not so surprising. I don't see any obvious reason why increasing resolution would automatically lead to substantial changes, nor why these changes would show up as improvements in large-scale features when compared to observations (it mainly depends on your luck!). The situation might be different if the resolution was high enough to explicitly resolve deep convection, but even the finer resolution considered here (40 km) is far too coarse for that.

It is correct that the resolution is still too large to explicitly resolve deep convection. We were hoping that the high resolution version would be closer to the reality due to the improved subgrid variability in the surface parameters, especially the SSTs, which is the main driver of convection in the tropics. However, we found that, in some cases, the estimates of heating rates in the high resolution version are even further away from the satellite estimates and we thought it would be worth documenting this.

11. lines 211, 219: The overly positive LW CRH in the upper troposphere is a curious feature. It is suggested that this is due to underestimated ice water content, which leads to underestimated cloud top LW cooling. This does not however explain why the modelled LW CRH is positive (middle column of Fig. 5). One possibility is that due to underestimated cloud fraction in the midtroposphere, too much upwelling LW radiation from lower levels reaches the high clouds, leading to LW heating.

This could very well be the case and is something worth mentioning in the discussion.

12. line 219: Regarding the large underestimate of IWC in EC-Earth compared to the satellite observations, I am wondering if precipitating ice is included in the IWC in the latter. In models it is generally not. There are also other potential reasons that could make the satellite profiles and the model profiles in Fig. 6 not to be fully compatible. For example, there is no satellite simulator in the model. Also, do the EC-Earth cloud fraction and cloud water fields include convective clouds?

Following the reviewer suggestion, we used different flags in the CloudSat data to filter precipitation. Fig. 6 is revised in order to show cloud water estimates with and without the precipitation. We see that, depending on whether we consider precipitation contaminated profiles or not, the models either underestimate or overestimate the cloud water, both in the liquid and ice phases. Cloud Fraction is already matched with the satellite overpass time, so it is a fair comparison. EC-Earth cloud fraction and cloud water fields analysed here include convective clouds.

13. In Figures 5 and 6, the results seem much the same for all seasons. Furthermore, in the text on lines 201–223, the seasonal differences are not discused at all. So why not simply show the annual-mean values? This would also allow combining Figs. 5 and 6 into one figure with six panels.

We agree. The figures and the associated text are revised in the revised manuscript.

14. line 247: this should be "radiative heating and cooling rate anomalies"?

Yes, this is corrected in the revised manuscript.

15. line 251: "a strong cooling (-0.75 K day ) above 10 km during ENSOP". I cannot find this large CRH anomaly over the Atlantic in Fig. 8, and at least it is not representative of the Atlantic region in general. Please also check that the other numerical values given in the text in Section 3.2.1 are consistent with Figs. 7 and 8.

We have checked the numerical values in the section to make sure that they are consistent with the Figures.

16. lines 253–266 and Figs. 9 and 10. The comparison of ENSO-related cloud radiative heating anomalies between different versions of EC-Earth seems largely useless, owing to the low statistical significance of the results. The low significance itself is not surprising considering your small sample size. I recommend to eliminate this part of the manuscript.

We agree. Given the very small differences, we have used the results from only one model version and mentioned the results from the other version only in the text.

17. line 267: The section title "Nino 3.4 region" is misleading, when you discuss separately Nino 3 and Nino 4 but not Nino 3.4.

True, we have changed the title in the revised version.

18. caption of Fig. 13: This should be "Cloud water content and cloud fraction anomalies".

Yes, this it is corrected.

19. line 292: "half of that" should be "twice that"?

Corrected. Thanks for pointing it out.

20. line 304: It would be relevant to comment on how your findings compare with those seen in the multimodel study of Cesana et al. (2019).

A brief note on such a comparison is added in the revised version.

21. line 317: Do you mean "two noticeable differences in net CRH"?

We agree that it is clearer to specify the "… net CRH" and have therefore added this to the sentence.

**Technical and language corrections**

1. Figures 2, 5, 6, 8, 11, 12, 13, and 14, and especially their labels, are painfully small to read. Please enlarge them. Too small figures are a sure method to make a reader (and a reviewer) förbannad.

2. The choice of colours in Figs. 5, 6, 11, 12, 13, and 14 is not good. It requires effort to distinguish SAT (a colder shade of purple) from E3PH (a warmer shade of purple/red). Please use clearly distinguishable colours.

The last thing we would like is to make the reviewer förbannad. Hopefully, the figures in the improved manuscript are up to the reviewers standard.

3. line 284: Replace "excessive" with "extensive". "Excessive" implies that there is too much cloudiness in the observations.

Excessive is replaced with extensive in the new manuscript.

---

## Author Comment (AC3) · 31 Mar 2021

**Response to Reviewer #2:**

We thank the reviewer for the constructive criticism that led to improvements in the manuscript. Following the suggestions from the both reviewers, we have carried out a major revision. The discussions and readability are improved and the redundancies in figures are removed. Among other things, Figs. 4, 5, 6, 8, and 11 are revised, while Figs. 9 and 10 are removed to declutter the discussion. Please note that few of the reviewer comments may not be valid anymore due to the revision of either the underlying figure or the text.

Please find below the point by point reply to your comments.

This study evaluates the performance of various versions of the EC-Earth model against satellite retrievals from CloudSat/Calipso focusing on the clouds and cloud radiative heating rates (CRH). The authors evaluate the intraseasonal and interannual variability of CRH in the tropics. This study is useful and would be a valuable contrition to the related and increasing body of literature that deals with the coupling between the circulation and CRH. However, the authors mostly illustrate a large amount of results without providing enough insights and deeper analysis of the results. Additionally, my impression was that the authors did not properly illustrate the gained knowledge from this study (potentially by showing many results without a clear story?) and did not communicate the main key points of this study to the reader. Another general remark could be the fact that the comparison among the different model versions was not giving any extra information to the paper and thus the authors should reconsider to focus on a specific model version and try to provide more deeper analysis and explanations for the differences between the EC-Earth model and the observations. As a final major remark, the readability of the vertical profiles was very bad and the authors should reconsider using more distinguishable colors.

We thank the reviewer for recognizing the potential of the results. We understand that the main message probably got cluttered in the way the analysis was presented. The other reviewer also raised similar concerns. We have revised the manuscript, considering the comments from both the reviewers, and improved the presentation, clarity and figure quality.

1) There is a confusing part in this study that comes from the fact that the authors do not clearly explain the method that they use to compare the models with the observations. Is it monthly mean, 3-hourly data that the authors use to compare the simulated CRH with the observed ones?

The model output is written out every three hours. From these results, we interpolate to the local time 13:30 to match the satellite overpass. All mean values (monthly /ENSO) are then based on these interpolated data. We have updated the paragraph starting on line 115 to "Note that the satellite simulator for evaluating heating rates in EC-Earth is currently not available. CloudSat and CALIPSO pass the equator at roughly 13:30 local time during daytime, so the model results are linearly interpolated from the two nearest output times (3–hourly data) to fit the satellite overpass time. This time interpolation together with the ability of the active satellite instruments to detect thin clouds reduces the need for satellite simulator otherwise commonly used for passive instruments (see, e.g. Pincus et al., 2012). All means (monthly/ENSO) in this study are based on these interpolated data."

2) Another general remark is that the authors in the results section mostly describe the figures without conceptually connecting the key points of the study and without providing more insights and explanations.

We have added more discussions and potential reasoning behind the observed biases in the revised version. The main aim of the manuscript is to compare/evaluate the CRH to understand if the EC-EARTH versions are able to simulate them well to a first order and which version is closest to the observations. Therefore, we had focused on documenting these biases in CRH and the relevant cloud properties, avoiding to speculate too much on the effects of underlying parameterizations (without doing the required idealized experiments). We hope that the new additions relating CRH biases to cloud properties and other potential factors (e.g. surface parameters) would provide more information in this context.

3) The authors illustrate the changes in CRH across seasons and different ENSO phases, however they do not illustrate the gained knowledge from this changes and do not properly communicate the key message from these comparisons in the context of previous studies.

We would very much like to discuss our results in the context of previous studies. However, to our knowledge, there is no previous study that specifically evaluates the vertical structure of CRH during ENSO phases in a climate model using combined CloudSat+CALIPSO data. Almost all of the previous studies have focused on investigating the top of the atmosphere cloud forcing during ENSO in different contexts, such as understanding cloud feedbacks. The main message from our evaluation of EC-EARTH is that all model versions capture the shift in meridional CRH associated with the changes in Walker circulation during the positive and negative phases of the ENSO. However, we see similar underestimations/overestimations in the vertical structure of CRH in the model climatology compared to the observations. This means that the basic description of cloud processes has to be improved first in the model before studying the general role of cloud radiative effects in the ENSO variability, especially related to cloud feedbacks.
We have clarified this in the revised version.

4) The readability of the figures is really bad (particularly those that include vertical profiles), thus more distinguishable colors should be used.

The figures are updated with clearer colours and larger fonts.

Minor comments:
Line 3: Typo: coupling

Corrected.

Line 148: Typo: analysis

This is now changed.

Line 159: It could be easier for the reader if the figures and the text had different longitude values to something like 45E, 90E, 90W, 45W.

Please note that we have shown plots from 0E to 360E instead of 180W-180E in order to have Pacific Ocean at the center of the plot in order to highlight the variability associated with ENSO.

Figure 2 (e-h): To make the comparison easier since the differences are larger than 1K/day wouldn't it make more sense to use the same colorbar as in panels (a-d) so that the reader can compare the differences compared to the absolute values?

We agree and have changed the colourbar in the revised manuscript.

Line 179: Do the authors refer to the temperature tendency due to convective parameterizations or the CRH? It should be properly phrased here.

It is the CRH. Clarified in the revised text.

Line 180-181: Since the model overestimates the magnitude of the CRH over both convectively active and stratocumulus regions is there a reason to separate them by using "but"?

Thanks. It is corrected.

Lines 183-184: Any potential explanation for that?

It is explained in Section 3.1.2 that discusses meridionally and zonally averaged differences. "Figure 6 shows that the models underestimate the cloud fraction as well as cloud liquid and ice water content in the lower and middle parts of the troposphere. This underestimate results in suppressed shortwave heating and longwave cooling in the middle troposphere in the models, thus, at least partly, explaining the differences observed in Figure 5."

Lines 184-186: This is confusing. Isn't it the case that panels (e-h) in Figure 2 show monthly mean differences between the model and the observations? See major comment 1

Figure 2 shows the monthly mean differences for the local time 13:30. Since the satellite only covers this time we interpolate the model results from the nearest 3-hourly data to the local time 13:30 for each day, then we calculate the monthly mean from these interpolated results. Hopefully this is clearer in the new manuscript.

Lines 189-190: It is not obvious that the convection is strong. This is not shown anywhere. Moreover, the authors do not provide enough evidence when is the dry and the wet season for a location. Maybe show this somewhere?

We have added references and updated the text in the revised manuscript.

Lines 187-199: Any further insights about these differences?

Two main factors can potentially explain this behaviour. The convection is strongly influenced by the surface temperature in the models. By improving the spatial resolution, the subgrid-scale variability in the SSTs is better represented and the modelled convection is more sensitive to the SSTs (compared to coarser grid values), leading to an even stronger hydrological cycle in the high-resolution version. Such behaviour is also seen in the regional climate models. Secondly, following the HighResMIP (High-Resolution Model Intercomparison Project) protocol, the same tuning is applied to the higher resolution version as well which may not be optimal and results in clouds that are too thick/include too much cloud water/live too long.
We have updated the discussion on this in the revised manuscript.

Lines 204-205: Isn't it the case that the magnitude in the lower troposphere (below 2 km) is smaller across the model versions, thus underestimating the magnitude of the cooling?

Yes, it is clarified in the new version.

Lines 210-212: Any insights about these differences? Could you potentially explain why the models simulate the maximum heating higher that in the observations? Moreover, the different model versions seems not to be sensitive to CRH. Is there any explanation why these differences are so small?

These differences are explained in response to the earlier comment.
There are a couple of factors that can cause the maximum heating in the models to be higher than in the observations. To begin with, the models heavily underestimate the cloud fraction between 5 to 12km altitude. These are typically deep convective and nimbostratus clouds that produce substantial cloud top cooling in the layers above (12-14 km). This cooling in the uppermost troposphere is missing in the models (Fig. 5). Instead, the clouds are predominantly located in the 12-14 km region in the models. The net effect of this is that the altitudes where observations show strong cloud top cooling are also the regions where models could have either heating or suppressed cooling, resulting in the maximum heating higher than the observations. This discussion is added in the revised manuscript.

The Primavera model is based on an early version of the model used for CMIP6 and differences between the modes are mainly in the surface albedo scheme (see Section 2.1). Therefore the large differences in the cloud radiative heating between these two different versions are not expected. However, these versions have not been evaluated before, therefore we believe it is a valuable comparison. CMIP6 version is the most state-of-the art, while PRIMAVERA versions are useful to investigate the impact of spatial resolution.

Figure 6: It might be better to use letters to show which panels show IWC, LWC and CF.

This is added in the new version.

Lines 218-220: This is somehow confusing. Despite CF differences in the upper troposphere are small, the models seems to underestimate IWC compared to the observations and and still producing stronger CRH than the observations. Could you explain
why is this the case?

The underestimation of cloud fraction and cloud liquid water path in the middle and lower troposphere (Fig. 6) allows more LW radiation to reach the upper layers, where it gets trapped and absorbed. Due to lower ice water in the upper layers, the resulting LW cooling is also lower at the cloud tops. This is now explained in the revised version.

Lines 214-215: This underestimation of CRH is mainly evident in the mid-troposphere. However, in the lower troposphere CRH agree more despite the fact that cloud properties differ substantially. Any explanation regarding that?

There are also disagreements in the lower troposphere, but they are not as strong as in the middle and upper troposphere (Fig. 5). When we look into the individual SW and LW heating components of the total CRH, we see that the opposite biases in these components lead to the lower net biases in the total CRH in the lower troposphere.

Lines 244-253: Could the authors provide more insights that would explain these differences?

The nature of these differences is similar to the ones reported in Figs. 2-4. Here as well we notice that the models underestimate middle and upper tropospheric cloud fraction (between 5 and 10 km), underestimating heating and cooling in the ENSO positive and negative phases respectively. This is explained more clearly in the revised version.

Figure 7: As before, either adjust the colorbar to make the comparison more straightforward or mention in the figure caption the different colorbar for the differences. Additionally, in the figure caption it would be more appropriate to mention the satellite
(CloudSat/Calipso) instead of mentioning CRH from satellite.

We have updated the colourbar and caption in the revised manuscript.

Lines 259-260: Could the authors providence more evidence why is this happening?

This comment is not applicable anymore. Figs. 9 and 10 are removed in the revised manuscript to avoid redundant information.

Line 262: Why the differences are mostly evident over the ENSO region? Is this consistent with previous modeling studies? What could be the reason behind these differences?

Due to the removal of Figs 9 and 10, this section is now revised and the differences are explained.

Line 274: Which figure shows that? Maybe add the figure at the end of the phrase?
Figure 11. This is clearer in the new manuscript.

Line 282: As before, which figure shows that?
Figure 11. This is clearer in the new manuscript.

Lines 295-298: Maybe use the same wording for anomalies? Using anomalies and changes in the same phrase might be confusing for the reader.

This is corrected in the revised manuscript.

---

## Referee Report (RR1)

**General comments**

The authors have addressed adequately most of the concerns that I raised in my original review. There is one point I disagree on: adding the SAT-WP curves to Fig. 7 does not actually seem very meaningful (see the specific comment 6). My other comments are all very minor and mainly related to "fine-tuning" the presentation.

**Specific comments**

1. lines 62-65: While Cesana et al. (2019a) is a relevant paper, it would be even more important to discuss briefly Cesana et al. (2019b) here (i.e., the heating rate paper). This is what I suggested in my original review.

2. lines 75–86: The flow of the text would be smoother, if you first described what EC-Earth v3 is, and how it is related to the CMIP5 versions EC-Earth v 2.3 — and only after that, start discussing the different versions and resolutions used for EC-Earth v3 in this study. A simple reorganization of the text would do the trick: move the contents of your current 2nd paragraph (lines 81–86) right after the first sentence ("The atmospheric model..."). Then, in a new paragraph, continue with the text on lines 76–80.

3. lines 125–129: I think it would be useful to say here explicitly that the use of data at roughly 13:30 local time emphasizes the relative role of SW heating rates as compared with LW heating rates. It is not self-evident that the readers will note this point otherwise.

4. Caption of Fig. 5: "averaged over the whole season" should probably be "averaged over all seasons" or "averaged over the annual cycle".

5. Caption of Fig. 6, 3rd line: "read" should be "spread"?

6. lines 250–255: I don't think that the statement "the cloud water content from the models does not include the contribution from precipitating clouds" characterizes the situation accurately (literally, this would mean that whenever there is precipitation, the LWC and IWC diagnosed by the model would be zero — which is certainly not the case!). Rather, the modeled LWC and IWC always include cloud water and cloud ice (i.e., cloud droplets and ice crystals small enough to stay in the cloud), whether or not the clouds are precipitating, but presumably

not rain and snow (i.e., water drops and ice particles large enough to fall out of the cloud due to the effects of gravity). Therefore, screening out entire columns of CloudSat/CALIPSO data when there is some precipitation does not provide a meaningful comparison with model data – it probably eliminates a large part of the condensed/frozen water that would be diagnosed as LWC and IWC by the model.

Therefore, I recommend eliminating the "SAT-WP" curves from Fig. 7. Just mention in the text that a potential reason contributing to the underestimate by the model is that the modeled LWC and IWC do not include rain and snow.

7. Caption of Fig. 12, 1st line: Add "anomalies" after "Cloud water content and cloud fraction."

8. lines 293, 294, 309 and 310: replace "models" with "model". You discuss the results for only one model version in this section.

---

## Referee Report (RR2)

Review comments
for GMD paper 2020-277

I would like to thank the authors for substantially revising the manuscript and improving the overall structure. In the revised version, the authors address all the points mentioned during the first revision round of the manuscript and now the overall presentation and discussion of the results is improved in the revised version. Furthermore, the authors improved substantially the communicated messages of their study. As I mentioned in the first revision of the manuscript, this study is very useful and would be a valuable contrition to the increasing body of literature that deals with the coupling between the circulation and cloud-radiative heating.

Thus, I recommend this paper for publication, and I briefly mention some minor comments for the authors below.

Minor comments:

Lines 55-68: Maybe worth mentioning also a similar recent study of Voigt et al., 2019 (https://doi.org/10.1175/JCLI-D-18-0810.1) where different climate models are compared with CloudSat/CALIPSO observations, as well as Zelinka et al., 2018 (https://doi.org/10.1175/JCLI-D-18-0114.1) where also multiple climate models are compared to CloudSat/CALIPSO retrievals?

Figures 2, 5, 9 could be slightly wider because in the current form they look smaller than the others.

---

## Author Response (AR2)

**Response to Reviewers**

We would like to thank the reviewers for taking the time to revise our manuscript and give feedback that has helped to improve the manuscript.
Please find below our point by point reply to your comments.

**Response to Reviewer #1**:

**General comments**
The authors have addressed adequately most of the concerns that I raised in my original review. There is one point I disagree on: adding the SAT-WP curves to Fig. 7 does not actually seem very meaningful (see the specific comment 6). My other comments are all very minor and mainly related to "fine-tuning" the presentation.

**Specific comments**
1. lines 62-65: While Cesana et al. (2019a) is a relevant paper, it would be even more important to discuss briefly Cesana et al. (2019b) here (i.e., the heating rate paper). This is what I suggested in my original review.

We have now changed the discussion to Cesana et al. (2019b)

2. lines 75–86: The flow of the text would be smoother, if you first described what EC-Earth v3 is, and how it is related to the CMIP5 versions EC-Earth v 2.3 — and only after that, start discussing the different versions and resolutions used for EC-Earth v3 in this study. A simple reorganization of the text would do the trick:
move the contents of your current 2nd paragraph (lines 81–86) right after the first sentence ("The atmospheric model. . . "). Then, in a new paragraph, continue with the text on lines 76–80.

The text is rearranged.

3. lines 125–129: I think it would be useful to say here explicitly that the use of data at roughly 13:30 local time emphasizes the relative role of SW heating rates as compared with LW heating rates. It is not self-evident that the readers will note this point otherwise.

We have added a sentence about this at the end of the paragraph.
"However, when using data from early afternoon (~13:30), when the incoming shortwave radiation is close to its peak, the contribution of shortwave radiation to the total heating is emphasised."

4. Caption of Fig. 5: "averaged over the whole season" should probably be "averaged over all seasons" or "averaged over the annual cycle".

"all seasons" are now used in the caption for Figure 5-7

5. Caption of Fig. 6, 3rd line: "read" should be "spread"?

Yes

6. lines 250–255: I don't think that the statement "the cloud water content from the models does not include the contribution from precipitating clouds" characterizes the situation accurately (literally, this would mean that whenever there is precipitation, the LWC and IWC diagnosed by the model would be zero — which is certainly not the case!). Rather, the modeled LWC and IWC always include cloud water and cloud ice (i.e., cloud droplets and ice crystals small enough to stay in the cloud), whether or not the clouds are precipitating, but presumably not rain and snow (i.e., water drops and ice particles large enough to fall out of the cloud due to the effects of gravity). Therefore, screening out entire columns of CloudSat/CALIPSO data when there is some precipitation does not provide a meaningful comparison with model data – it probably eliminates a large part of the condensed/frozen water that would be diagnosed as LWC and IWC by the model. Therefore, I recommend eliminating the "SAT-WP" curves from Fig. 7. Just mention in the text that a potential reason contributing to the underestimate by the model is that the modeled LWC and IWC do not include rain and snow.

Thanks for the suggestion. We agree. We have removed the SAT-WP from Figure 7 and replaced the paragraph (lines 250-255) with the sentence (at line 239-240):

"An explanation for the models underestimating liquid and ice water content is that the models do not include the contribution from rain and snow."

7. Caption of Fig. 12, 1st line: Add "anomalies" after "Cloud water content and cloud fraction."

"anomalies" is added.

8. lines 293, 294, 309 and 310: replace "models" with "model". You discuss the results for only one model version in this section.

Absolutely, the "s" is removed.

**Response to Reviewer #2:**

I would like to thank the authors for substantially revising the manuscript and improving the overall structure. In the revised version, the authors address all the points mentioned during the first revision round of the manuscript and now the overall presentation and discussion of the results is improved in the revised version. Furthermore, the authors improved substantially the communicated messages of their study. As I mentioned in the first revision of the manuscript, this study is very useful and would be a valuable contrition to the increasing body of literature that deals with the coupling between the circulation and cloud-radiative heating. Thus, I recommend this paper for publication, and I briefly mention some minor comments for the authors below.

**Minor comments:**
Lines 55-68: Maybe worth mentioning also a similar recent study of Voigt et al., 2019 (https://doi.org/10.1175/JCLI-D-18-0810.1) where different climate models are compared with CloudSat/CALIPSO observations, as well as Zelinka et al., 2018 (https://doi.org/10.1175/JCLI-D-18-0114.1) where also multiple climate models are compared to CloudSat/CALIPSO retrievals?

Thanks for the suggesting these relevant references. We have cited them in the revised version.

Figures 2, 5, 9 could be slightly wider because in the current form they look smaller than the others.

The width of the figures is regulated by the GMDs latex style guide (not exceeding 12 cm, which they currently are).